# Pretreatment of Anthocyanin from the Fruit of *Vitis coignetiae Pulliat* Acts as a Potent Inhibitor of TNF-α Effect by Inhibiting NF-κB-Regulated Genes in Human Breast Cancer Cells

**DOI:** 10.3390/molecules25102396

**Published:** 2020-05-21

**Authors:** Anjugam Paramanantham, Min Jeong Kim, Eun Joo Jung, Arulkumar Nagappan, Jeong Won Yun, Hye Jung Kim, Sung Chul Shin, Gon Sup Kim, Won Sup Lee

**Affiliations:** 1Departments of Internal Medicine, Institute of Health Sciences, Gyeongsang National University Hospital, Gyeongsang National University School of Medicine, Jinju 660-702, Korea; anju.udhay@gmail.com (A.P.); bokdae@hanmail.net (M.J.K.); arulbiotechtnau@gmail.com (A.N.); potato-yun@hanmail.net (J.W.Y.); 2Research Institute of Life science and College of Veterinary Medicine, Gyeongsang National University, 501 Jinju-daero, Jinju 52828, Korea; 3Departments of Biochemistry, Institute of Health Sciences, Gyeongsang National University Hospital, Gyeongsang National University School of Medicine, Jinju 660-702, Korea; eunjoojung@gnu.ac.kr; 4Departments of Pharmacology, Institute of Health Sciences, Gyeongsang National University Hospital, Gyeongsang National University School of Medicine, Jinju 660-702, Korea; curlysookim@hanmail.net; 5Department of Chemistry, Research Institute of Life Science, Gyeongsang National University, Jinju 660-701, Korea; scshin@gsnu.ac.kr

**Keywords:** anthocyanins, *Vitis coignetiae Pulliat*, NF-κB, breast cancer

## Abstract

*Vitis coignetiae**Pulliat* (Meoru in Korea) has been used in Korean folk medicine for the treatment of inflammatory diseases and cancers. Evidence suggests that NF-κB activation is mainly involved in cancer cell proliferation, invasion, angiogenesis, and metastasis. TNF-α also enhances the inflammatory process in tumor development. Recently, flavonoids from plants have been reported to have inhibitory effects on NF-κB activities. We investigated the effects of anthocyanins extracted from the fruits of *Vitis coignetiae Pulliat* (AIM, anthocyanins isolated from Meoru (AIM)) on TNF-α-induced NF-κB activities in MCF-7 human breast cancer cells and the molecules involved in AIM-induced anti-cancer effects, especially on cancer metastasis. We performed cell viability assay, gelatin zymography, invasion assay, and western blot analysis to unravel the anti-NF-κB activity of AIMs on MCF-7 cells. AIM suppressed the TNF-α effects on the NF-κB-regulated proteins involved in cancer cell proliferation (COX-2, C-myc), invasion, and angiogenesis (MMP-2, MMP9, ICAM-1, and VEGF). AIM also increased the expression of E-cadherin, which is one of the hallmarks of the epithelial-mesenchymal transition (EMT) process. In conclusion, this study demonstrates that the anthocyanins isolated from the fruits of *Vitis coignetiae Pulliat* acts as an inhibitor of TNF-α induced NF-κB activation, and subsequent downstream molecules involved in cancer proliferation, invasion, adhesion, angiogenesis, and thus have anti-metastatic activities in MCF-7 breast cancer cells.

## 1. Introduction

Cancer is one of the fatal diseases in most countries. In female, the most frequently diagnosed type of cancer is breast cancer, which still prevails as the prime cause of high female fatality rate. It is reported that around 23% of the cancer types are found to be breast cancer and 14% of female fatality have been reported due to breast cancer [1]. Hence, extensive research in this field is required in order to overcome both economical and psychological burden [2]. Metastatic activity is one of the leading causes of death in breast cancer; more than 50% of the cases succumb to this disease due to the metastatic activity. Therefore, it is crucial to address the metastatic activity of cancer progression in breast cancer [3].

Tumor metastasis occurs through an intricate series of events including cell adhesion, invasion, proliferation, and vessel formation [4]. Basement membrane degradation and stromal extracellular matrix (ECM) deterioration are essential for the invasion and metastasis of malignant cells. Therefore, the control of metastatic activity is crucial for the management of breast cancer. Conventional systemic chemotherapy is a mainstay treatment for inoperable breast cancer. Although the side effects of chemotherapy are relatively high and serious in elderly patients, they result in death in some patients. Therefore, the demand for less toxic agents for breast cancer is tremendous. Recent studies have shown that dietary agents have high anti-cancer activities and enhanced safe drug delivery.

To treat various diseases like cancer and inflammatory diseases, anthocyanins isolated from *Vitis coignetiae Pulliat* (Meoru in Korea) are used as a Korean folk medicine. The fruits are dark red in color, which contains an abundance of anthocyanins belonging to a class of flavonoids. Recently, the anti-cancer activities of anthocyanins have been demonstrated regarding anti-angiogenesis and cancer invasion [5,6]. We previously suggested that the anthocyanins (AIM) isolated from Meoru (*Vitis coignetiae Pulliat)* may suppress cancer invasion through suppression of the NF-κB pathway in HT-29 human colon cancer cells [7]. Fatal cancer cells are highly invasive and have high metastatic activity, which has been controlled by Nf-kB through regulating the transcriptional activity of matrix metalloproteinase (MMP) and angiogenic enzymes [8]. Natural polyphenols have been shown to regulate the expression of a number of genes involved in tumorigenesis as well as cancer metastasis [9,10,11]. These include anti-apoptosis genes such as TRAF, bcl-2, cyclin D1, c-Myc, and cIAPs [12,13]. The inflammatory cytokines like TNF-α (tumor necrosis factor) and IL-1β (InterLeukin-1β) are mainly regulated by Nf-κB, an essential transcription factor, which in turn activates MMP-9 and COX-2 [9,10,14]; thus, several natural phytochemicals are able to suppress NF-κB activation, resulting in suppression of tumorigenesis and metastasis. We have previously observed that AIM showed anti-cancer effects on hepatocellular cancer [15] and colon cancer cells [7] by suppressing NF-κB. However, AIM influence on NF-κB-regulated proteins in breast cancer cells has not been much explored. TNF-α can induce cancer cell death when treated in high concentration [16], but in low concentration, it promotes metastasis [17,18]. Here, we investigated the effects of TNF-α pretreated with AIM on NF-κB-regulated proteins in MCF-7 cells, focusing on cancer metastasis involved in cancer invasion, adhesion, and angiogenesis.

## 2. Results

### 2.1. Anthocyanins Isolated from Meoru (AIM) Inhibited the Cell Proliferation, Tumor Necrosis Factor (TNF)-Augmented Cell Adhesion of MCF–7 Cells

We assessed the effects of AIM on the growth of MCF–7 cells at different time intervals (24 h, 48 h, and 72 h) after treatment. The MTT assay revealed that AIM suppressed the proliferation of MCF–7 cells in a dose-dependent manner at 48 h and 72 h (Figure 1B). However, AIM showed no effect on MCF-7 cells at 24 h treatment. AIM strongly inhibited cell proliferation at the concentration of 400 μg/mL when compared to the controls in 48 h and 72 h. Furthermore, we investigated the effect of AIM on the adhesion of MCF-7 cells to human umbilical vein endothelial cells (ECs) at the lower concentration (10–200 μg/mL) of AIM. The adhesion assay revealed that AIM significantly inhibited TNF-augmented cancer cell adhesion of MCF–7 cells in a dose-dependent manner (Figure 2B). Taken together, these results strongly suggest that AIM has anti-cancer properties on cancer proliferation and the cell adhesion of MCF–7 cells. Gelatin zymography revealed MMP-2 and MMP-9 were inhibited in a dose dependent manner (Figure 2A). Western blot analysis also revealed that AIM inhibited TNF-α induced effect by inhibiting MMP-2 and MMP-9in a dose dependent manner (Figure 2C). Complete inhibition of MMP-2 and MMP-9 was observed in both gelatin zymography and western blot analysis.

### 2.2. Pre-Treatment of AIM Inhibits TNF-α Induced Metastasis Activity

The first step of tumor invasion is proteolytic digestion of the extracellular matrix (ECM) and the basement membrane. In particular, MMP-2 (gelatinase-A) and MMP-9 (gelatinase-B) play important roles in the proteolytic digestion of the basement membrane by degrading type IV collagen, and their expressions are regulated by NF-κB [19]. Therefore, we assessed the effect of TNF-α, a stimulant of NF-κB pre-treated with AIM on breast cancer cells. The Matrigel test revealed that TNF-α significantly increased cancer cell invasion, and that AIM also significantly reduced cancer cell invasion augmented by TNF-α (Figure 3A). In addition, to confirm the involvement of NF-κB-regulated proteins MMP-2 and MMP-9 expression, we measured the gelatinolytic activity of MMP-2 and MMP-9 secreted from MCF-7 cells. In contrast, TNF-α did not increase gelatinolytic activity of MMP-2 and MMP-9 secreted from MCF-7 cells. However, with or without the treatment of TNF-α, AIM completely inhibited the gelatinolytic activity of MMP-2 and MMP-9 (Figure 3B). These findings suggest that AIM inhibited cancer invasion by inhibiting the gelatinolytic activity of MMP-2 and MMP-9.

### 2.3. TNF-α Induced Effect Was Reversed with the Treatment of AIM Prior by Suppression of NF-κB Regulated Proteins Involved in Proliferation, Invasion, and Angiogenesis

We previously observed that AIM inhibited NF-κB activation in hepatocellular cancer [15] and colon cancer cells [7]. Several anti-cancer pathways like proliferation, invasion, and angiogenesis are activated after the activation of NF-κB. COX-2 and C-myc are overexpressed in different types of cancers and mediate cancer cell proliferation [20]. The roles of MMP-2, MMP-9, ICAM-1, and VEGF are well known in the invasion and angiogenesis of cancer [21]. The suppression of E-cadherin is one of the indications of the development of epithelial-mesenchymal transition (EMT) and metastatic cancer. The suppression of E-cadherin also promotes the transition from an epithelial to mesenchymal state. Lower expression of E-cadherin directly corresponds to the increased activation of β-catenin, EMT expression, and progression of cancer. [22]. All of these genes are known to be regulated by NF-κB [23,24]. Additionally, we investigated the effect of AIM on these proteins. Western blot analysis revealed that TNF-α increased the expression of cyclin D1, C-myc, COX-2, ICAM-1, and VEGF-2 in comparison with the AIM treated group. AIM suppressed TNF-α effects on the NF-κB-regulated proteins involved in cancer cell proliferation (COX-2, and C-myc), invasion, and angiogenesis (MMP-2 and MMP9, ICAM-1 and VEGF). AIM also increased the expression of E-cadherin, which is one of the hallmarks of the EMT process (Figure 4). These results suggest that AIM suppressed TNF-α-induced NF-κB-regulated proteins involved in proliferation, invasion, and angiogenesis in MCF-7 cells, and increased the expression of E-cadherin involved in EMT.

### 2.4. AIM Suppresses NF-κB Activity Partially Through Degradation of IκBα Phosphorylation

Western blot analysis and luciferase assay were performed to analyze the effect of AIM on NF-κB activity in MCF-7 cells. The luciferase assay revealed that the TNF-α induced NF-κB activity was inhibited by AIM. The results indicated that the NF-κB gene was transfected successfully into the cells (Figure 5A). The heterodimer of p50 and p65 translocates from the nucleus to cytoplasm after the activation of NF-κB; during resting conditions, the heterotrimer of p50, p65 and inhibitory κBα (IκBα) can be found in cytoplasm. AIM depleted the level of Nf-kB in cytoplasm, and nuclear translocation was also revealed by western blot analysis (Figure 5B). NF-κB activation leads to the degradation of IκBα through the phosphorylation of IκBα, followed by ubiquitination. In addition, we explored the AIM effect on TNF-α-induced IκBα phosphorylation. The western blot analysis clearly showed that the AIM suppressed IκBα phosphorylation, even in the presence of TNF-α.

## 3. Discussion

This study was designed to investigate the anti-cancer effects of AIM on NF-κB-regulated proteins and cellular responses induced by TNF-α in breast cancer cells. Here, we report that AIM acts as an inhibitor of the TNF-α-induced effect and inhibits the growth, adhesion, and invasion of MCF-7 cells. Furthermore, AIM inhibited the NF-κB-regulated proteins involved in cell proliferation, invasion, and angiogenesis. The activation of the NF-κB pathways is mainly involved in the inflammation, cancer cell proliferation, invasion, adhesion, and angiogenesis [23,24]. Edible berry juice containing plentiful anthocyanins and dietary anthocyanidin delphinidin have been reported to inhibit the NF-κB pathway [25,26]. In addition, NF-κB is known to play a significant role in cancer cell invasion and metastasis. We found that AIM suppressed NF-κB activation induced by TNF-α in MCF-7 cells, and the upregulation of NF-κB-regulated genes (COX-2; MMP-2, MMP-9, and VEGF) involved in cancer cell proliferation (COX-2, C-myc, Cyclin-D1), and invasion, adhesion, and angiogenesis (MMP-2, MMP-9, ICAM-1, VEGF). Our current findings were consistent with previous studies showing that few natural compounds inhibiting NF-κB activation have inhibitory effects on cancer cell proliferation and metastasis [26,27]. In addition, AIM showed inhibitory effects on the expression of MMP-2 and MMP-9 in MCF-7 cells, which is similar with the previous studies demonstrating the inhibitory effects of AIM on the expression of MMP-2 and MMP-9 in hepatocellular carcinoma cells [28].

Tumor invasion is the first step for the metastasis, and the invasion starts with proteolytic digestion of the extracellular matrix (ECM) and the basement membrane. Among the processes, degradation of type IV collagen is mainly carried out by MMP-2 (gelatinase-A) and MMP-9 (gelatinase-B) [19]. Therefore, these two proteins were accepted as targets for cancer progression because complete inhibition of invasion leads to the prevention of cancer metastasis. However, the AIM effects on NF-κB activity could be an indirect mechanism because AIM have anti-Akt activity as well as anti-EGFR [29]. These inhibitory effects on Akt or EGFR can contribute to the inhibitory effects of AIM on NF-κB activation.

The merit of this study is that we confirmed that AIM inhibited most of NF-κB-regulated proteins involved in cancer metastasis in human breast cancer cells. The limitation of this study is as follows, first, the detailed mechanisms for up-stream signaling for NF-κB activity of AIM were not fully elucidated. A previous report suggested that one of the components of AIM, delphinidin, inhibits the expression of kinase phospho-IKK, which is an upstream kinase of phospho-IkBα that is a direct inhibitor of NF-κB [30], and inhibits IL-1-induced NF-κB activation [31]. However, the expression of kinase phospho-IKK is also regulated by the PI3K-Akt pathway [32,33]. In this study, we did not answer why AIM for increased the cytoplasmic NF-κB and total NF-κB, and how AIM reduced the IkBa total protein level with the lack of IkBa phosphorylation. Regarding the reduction of IkBa with the lack of IkBa phosphorylation, IkB kinase-independent IkBα degradation has been reported [34]. Regarding these questions, further study is warranted. Despite previous reports mainly focused on the anti-cancer activities of AIM, here, we mainly focused on the inhibition of the TNF-α induced effect by AIM.

Second, we used TNF-α, one of the NF-κB stimulants to clearly show the effects of AIM on NF-κB activity and NF-κB-regulated proteins involved in cancer metastasis because base-line NF-κB activity of MCF-7 cell was low. The use of TNF-α appears to be artificial, but in the process of cancer metastasis, TNF-α or other cytokines from cancer and surrounding inflammatory cells play an important role in metastasis [35,36]. In addition, TNF-α is highly expressed in advanced cancer, and the TNF-α inhibitor shows anti-cancer effects by inhibiting cancer metastasis [36].

Finally, AIM is composed of more various anthocyanins than a single component of anthocyanins. We could not fully delineate which component of anthocyanins is the key molecule for this mechanism. Recently, a mixture of phytochemicals was evaluated on the anti-cancer effects as a single agent for cancer management. Therefore, AIM also may be accepted as single agent. Regarding the concentration of the AIM used in this study, it might seem high for in vivo studies and in normal cells, but in the adhesion study, no toxicity was observed to ECs and we have previously demonstrated its effects at several concentrations of AIM in in vivo studies [37].

In conclusion, this study suggests that AIM acts as an inhibitor of the TNF-α induced effect by the suppression of NF-κB-regulated gene expression in breast cancer cells. In addition, this study suggests that AIM has anti-metastasis effects because AIM suppressed the proliferation, adhesion of cancer cells to ECs, and invasion as well as the gene expression involving cell proliferation, invasion, and angiogenesis (Figure 6). Finally, this study provides evidence that AIM can act as a TNF-α inhibitor on human breast cancer.

## 4. Materials and Methods

### 4.1. Cell Culture and Chemicals

The breast cancer cell line MCF-7 acquired from ATCC (Rockville, MD, USA) were subcultured with Roswell Park Memorial Institute Medium (RPMI) 1640 media (Hyclone, Marlborough, MA, USA) containing 10% of heat inactivated (*v/v*) FBS (fetal bovine serum) (GIBCO BRL, Grand Island, NY, USA), 1 mM l-glutamine, 100 U/mL penicillin, and 100 μg/mL streptomycin at 37 °C in a humidified atmosphere of 95% air and 5% CO_2_. Human umbilical vein endothelial cells (EA.hy 926 cells, ECs) were obtained from ATCC and cultured in medium 199 (GIBCO BRL, Grand Island, NY, USA) supplemented with 20% FBS, 2 mM l-glutamine, 5 U/mL heparin, 100 IU/mL penicillin, 10 μg/mL streptomycin, and 50 μg/mL EC growth supplements. Antibodies against COX-2, MMP-2, MMP-9, VEGF, β-Catenin, Cyclin D1, C-myc, ICAM, E-cadherin, Survivin, IkB, p-IkB, and NF-κB were purchased from Santa Cruz Biotechnology (Santa Cruz, CA, USA). Antibody against β-actin was from Sigma (Beverly, MA, USA). Peroxidase-labeled donkey anti-rabbit and sheep anti-mouse immunoglobulin, and an enhanced chemiluminescence (ECL) kit were purchased from Amersham (Arlington Heights, IL, USA). All other chemicals not specifically cited here were purchased from Sigma Chemical Co. (St. Louis, MO, USA). We used the anthocyanins isolated from *Vitis coignetiae Pulliat* [8].

### 4.2. AIM Preparation

AIM was extracted from the fruits of Meoru. The well matured Meoru fruits were collected at Jiri Mountain, Republic of Korea. Purification and characterization of AIM (anthocyanins in Meoru) have been described previously [38]. AIM has the following composition: delphinidin-3,5-diglucoside:cyanidin-3,5-diglucoside:petunidin-3,5-diglucoside:delphinidin-3-glucoside:malvdin-3,5-diglucoside:peonidin-3,5-diglucoside:cyanidin-3-glucoside:petunidin-3-glucoside:peonidin-3- glucoside:malvidin-3-glucoside ¼ 1.0:0.5:3.4:28.1:6.4:6.4:4.2:22.5:4.9:22.5:5.0:22.6.

### 4.3. Cell Proliferation Assays

The MTT assay was performed to analyze the effect of AIM on the MCF-7 breast cancer cell line. MCF-7 cells were seeded with the seeding density of 5 × 10^4^ cells/ml in 24 well plates and treated with 0–400 µg/mL of AIM. After treatment, the cells were incubated for 24, 48, and 72 h at 37 °C in a CO_2_ incubator; subsequently, the cells were added to 50 μL of MTT solution (5 mg/mL in 1× PBS) and incubated for 3 h at 37 °C in a CO_2_ incubator. After incubation, the supernatant was removed and 200 μL of DMSO was added and kept in shaker for 15 min. The absorbance was measured at 570 nm on a microplate reader (Bio-Rad, Hercules, CA, USA).

### 4.4. Adhesion Assay

The cell-to-cell adhesion assay was performed as described [39]. In brief, human umbilical vein endothelial cells (ECs) and breast cancer cell line MCF-7 were treated with the indicated concentrations of AIM, followed by TNF-α stimulation for 6 h. Subsequently, MCF-7 cells with the seeding density of 7.5 × 10^5^ cells/mL was added to ECs. After a 30-minute incubation, the cell suspension was removed, and the cells were washed with 1× PBS three times. Then, the cells were photographed and counted under light microscope (CKX41 with a camera (Nikon, DS-U3)).

### 4.5. Cell Invasion Assay

The Transwell assay was performed to analyze the effect of AIM on cell invasion with or without TNF-α treatment The upper chamber of the Boyden chamber was coated with Matrigel (0.5 mg/mL) (BD Biosciences, San Jose, CA, USA) and incubated at 37 °C for 4 h. MCF-7 cells were cultured overnight in a serum free media. A total of 5 × 10^4^ cells/well of MCF-7 cells were added to the upper chamber after the solidification of Matrigel, with the serum free media with and without AIM (400 μg/mL) treatment followed by TNF-α (10 ng/ml) stimulation. RPMI media (500 µL) with 20% FBS was added to the lower chamber as a chemoattractant and incubated for 18 h at 37 °C in CO_2_ incubator. After incubation, the upper chamber was removed and washed with 1 × PBS. The lower part of the upper chamber was fixed with 10% formalin, followed by the 4′,6-diamidino-2-phenylindole (DAPI) treatment. After staining with DAPI, the invasive cells were counted manually using a fluorescent microscope.

### 4.6. Gelatin Zymography

To perform gelatin zymography, MCF-7 cells were seeded in a s6-well plate with the seeding density of 5 × 10^4^ cells/well followed by the treatment of AIM for 24 h and TNF-α for 6 h. The cells were incubated at 37 °C in a CO_2_ incubator. After incubation, the media were removed and resolved in 12% polyacrylamide gel containing gelatin (1 mg/mL). The gels were washed with 2.5% of Triton X-100 for 1 h and then incubated in activation buffer (50 mM Tris–HCl, pH 7.5, 10 mM CaCl_2_) for 16 h at 37 °C. After incubation in the activation buffer, the gels were stained with staining solution containing 10% glacial acetic acid, 30% methanol, and 1.5% Coomassie brilliant blue for 1 h. After washing, the gels revealed the white lysis zones indicating gelatin degradation, showing the status of MMP-9 and MMP-2.

### 4.7. Transfection

NF-κB-luciferase constructs were obtained from Dr. G. Koretzky (University of Pennsylvania). The construct was designed in a pGL3 basic luciferase expression vector with a consensus NF-κB binding sequence. Transient transfection was performed using Lipofectin (Gibco-BRL), according to the manufacturer’s protocol.

### 4.8. Luciferase Assay

The cells were grown in a 6-well plate with the seeding density of 5 × 10^4^ cells/well with or without AIM in the concentration of 400 µg/mL, followed by TNF-α stimulation for 6 h, and the cells were incubated at 37 °C in a CO_2_ incubator. After incubation, the cells were washed with 1 × PBS three times, and then lysed with the passive lysis buffer provided in the dual luciferase kit (Promega, Madison, WI, USA). According to the manufacturer’s protocols, the cells were assayed for luciferase activity using a TD-20/20 illuminometer. The ratio between Firefly and Renilla luciferase is expressed as a bar graph.

### 4.9. Western Blotting

The cells were grown in a 10-cm dish plate with the cell seeding density of 2 × 106 cells/plate. Then, the cells were treated with different concentrations of AIM and DMSO as a vehicle for 24 h and successive treatment of TNF-α, and then kept in 37 °C in a 5% CO_2_ incubator with different time points (0–24 h). After treatment, the cells were scraped with the floating cells and centrifuged for about 5 min at 2000 rpm. Then, the cells were washed with 1× PBS (pH of 7.4). To lyse the cells, buffer A, containing 10 mM 4-(2-hydroxyethyl)-1-piperazineethanesulfonic acid (HEPES) (pH 7.9), 1.5 mM MgCl2, 0.5 mM dithiothreitol (DTT), 5 μM leupeptin, 2 μM pepstatin A, 1 μM aprotinin, and 20 μM phenylmethylsulfonyl fluoride, was used. Then, they were thawed and frozen repeatedly. The cells were centrifuged at 2000 rpm for 15 min to separate the nuclear and cytoplasmic extract. The supernatant, which is a cytoplasmic extract, was removed and washed with buffer A once, followed by re-suspension with buffer B containing 10 mM Tris–Cl (pH 7.5), 0.5% deoxycholate, 1% Nonidet P-40, 5 mM EDTA, 0.5 mM DTT, 5 μM leupeptin, 2 μM pepstatin A, 1 μM aprotinin, and 20 μM phenylmethylsulfonyl fluoride. The supernatant suspended in buffer B was kept at 4 °C with shaking for 30 min and then centrifuged at 3000 rpm for 20 min. The supernatant fraction containing the nuclear proteins was collected. For isolation of the total cell extracts, cells were lysed in PRO-PREP protein extract solution purchased from iNtRON Biotechnology, Inc.( Dedham, MA, USA). For total protein isolation, the pellets were suspended in 2× sample buffer containing 100 mM of Tris-Cl (pH 6.8), 4% (*w/v*) sodium dodecyl sulfate (SDS), 0.2% (*w/v*) Bromophenol blue, and 200 mM of DTT (dithiothreitol) was added and heated at 100 °C for 10 min. Then, the concentrations of the cell lysate proteins were determined by means of the Bradford protein assay (Bio-Rad lab, Richmond, CA, USA) using bovine serum albumin as the standard. The protein (30 μg) was resolved by electrophoresis, electro transferred to a polyvinylidene difluoride membrane (Millipore, Bedford, MA, USA), and then incubated with primary antibodies, followed by secondary antibody conjugated to peroxidase. Blots were developed with an Enhanced chemiluminescence (ECL) detection system.

### 4.10. Statistics

All experiments were performed in triplicate and the results were expressed as a mean ± SD (standard deviation). We used the Student’s t-test for comparing two groups and Neuman–Keuls for more than three groups. * *p* < 0.05, ** *p* < 0.01 were considered significant. The densitometry analysis of the western blot gel pictures was done using ImageJ software (image processing and analysis in java) [40]

## Figures and Tables

**Figure 1 molecules-25-02396-f001:**
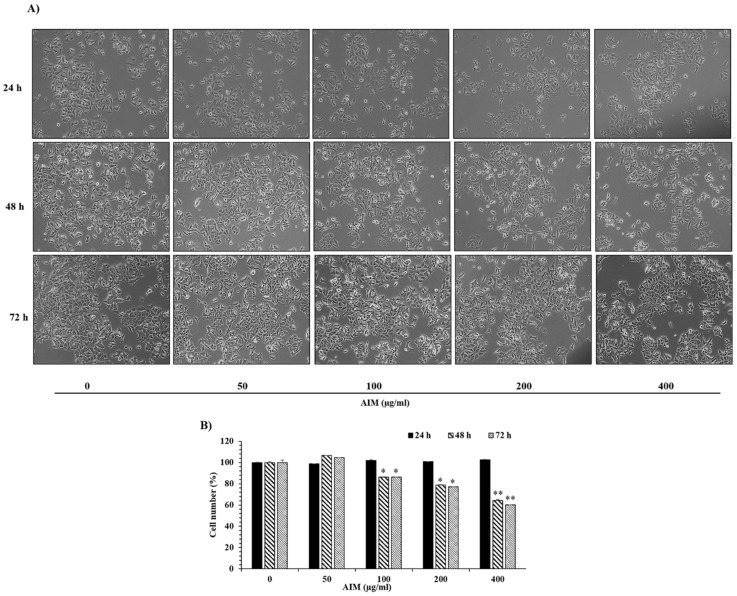
The inhibitory effects of anthocyanins isolated from Meoru (AIM) on cancer cell proliferation of MCF-7 breast cancer cells. (**A**) Morphological representation of MCF-7 cells with AIM treatment at different concentrations (0, 50, 100, 200, and 400 μg/mL) and time points (24 h, 48 h, and 72 h) were observed under light microscope (magnification, ×200; the length of scale bar, 50 μm). (**B**) Dose-dependent inhibitory effects of AIM on cell proliferation were assessed by the MTT assay. Cells were treated with indicated concentrations of AIM (0, 50, 100, 200, and 400 μg/mL) for 24 h, 48 h, and 72 h. Data are mean ±SD values from three independent experiments. * *p* < 0.05, ** *p* < 0.01 versus control.

**Figure 2 molecules-25-02396-f002:**
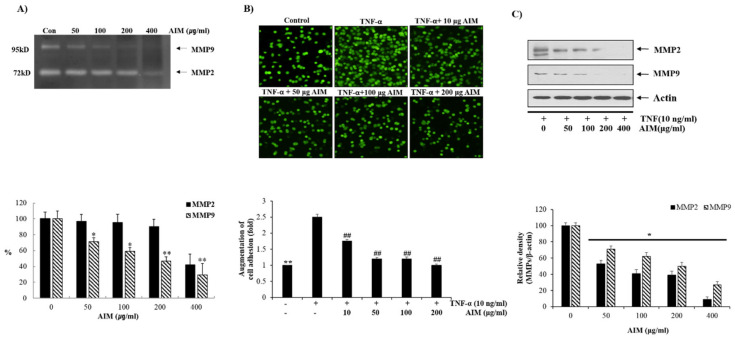
Inhibitory effects of AIM on cancer cell adhesion of MCF-7 Cells. (**A**) The cells were grown in 6-well plates with the seeding density of 5 × 10^4^ cells/well. MMP-2 and MMP-9 levels in gelatin zymography were assessed by densitometry. (Upper) The cells were treated with and without AIM for 24 h and the conditioned medium was taken to perform gelatin zymography. (Lower) MMP-2 and MMP-9 are expressed as a percentage of the activities against untreated cells. (**B**) The cells were seeded at the density of 5 × 10^4^ cells/mL. The cells were treated with indicated concentrations of AIM (0, 50, 100, and 200 μg/mL) for 24 h and the AIM effect on cancer cell invasion was assessed. For the group treated with AIM and TNF-α, cells were pre-treated with AIM (0, 10, 50, 100, and 200 μg/mL) for 1 h and then treated with TNF-α (10 ng/mL). ** *p* < 0.01 vs. the control group, ## *p* < 0.01 vs. the TNF-treated group. (**C**) Cells (5 × 10^4^ cells), either left untreated or pre-treated with AIM for 1 h, were exposed to TNF-α (10 ng/mL) for 24 h. (Upper) 30 μg of whole cell protein lysate were used for western blot analysis using antibodies of MMP-2 and MMP9. (Lower) Densitometry analysis of the data in western blot analysis by ImageJ software. The values were normalized against β-actin. ** *p* < 0.01 vs. the control group.

**Figure 3 molecules-25-02396-f003:**
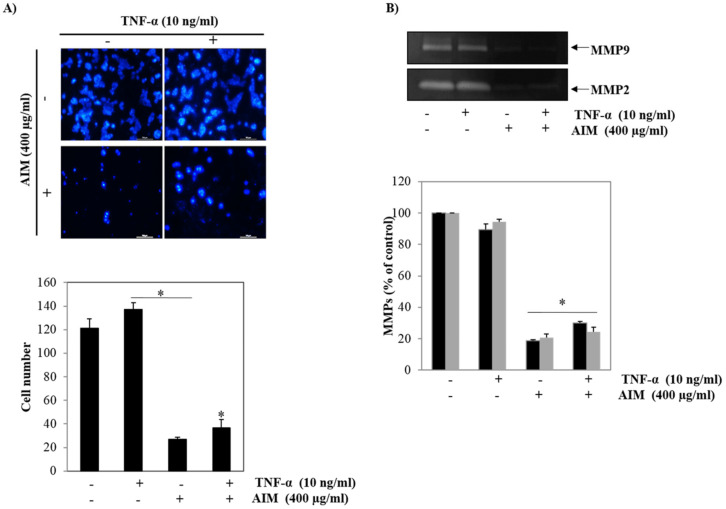
Inhibitory effects of AIM on the expression of TNF-treated MMP-2 and MMP-9 in MCF-7 cells. (**A**) AIM greatly influenced TNF-α -stimulated cell invasion of MCF-7 cells. (Upper) The Transwell invasion assay was performed using a Boyden chamber treated with or without TNF-α in the presence of AIM (400 μg/mL) at 37 °C for 24 h in a CO_2_ incubator. (Lower) Graphical representation of the number of invasive cells present were expressed in percentage. (**B**) MCF-7 cells were treated with TNF-α. (Upper) MMP-2 and MMP-9 protein secreted in the treated medium was used for the gelatin zymography analysis. (Lower) MMP-2 and MMP-9 enzyme activities were expressed in percentage and compared against the untreated cells. Data are the mean ± SD values of three independent experiments. * *p* < 0.05 versus the control.

**Figure 4 molecules-25-02396-f004:**
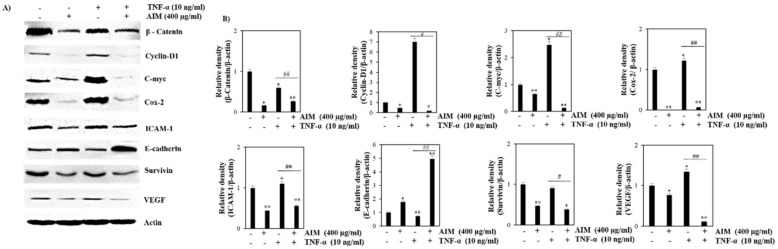
Effects of AIM and TNF-α induced AIM on NF-κB-regulated proteins involved in proliferation, invasion, and angiogenesis. MCF-7 cells were seeded with the seeding density of 5 × 10^4^ cells and pretreated AIM (400 μg/ml) for 1 h, followed by the treatment of TNF-α (10 ng/mL) for 24 h. The control cells were left untreated. The whole cell protein lysate was prepared and 30 μg of proteins were resolved in SDS-Polyacrylamide gels. (**A**) Western blot analysis of various NF-κB related proteins involved in cancer cell proliferation, invasion, and angiogenesis, (**B**) Densitometry analysis of the data in western blot analysis by ImageJ software. The values were normalized against β-actin. * *p* < 0.05, ** *p* < 0.01 vs. the control group, # *p* < 0.05, ## *p* < 0.01 vs. the TNF-α treated group.

**Figure 5 molecules-25-02396-f005:**
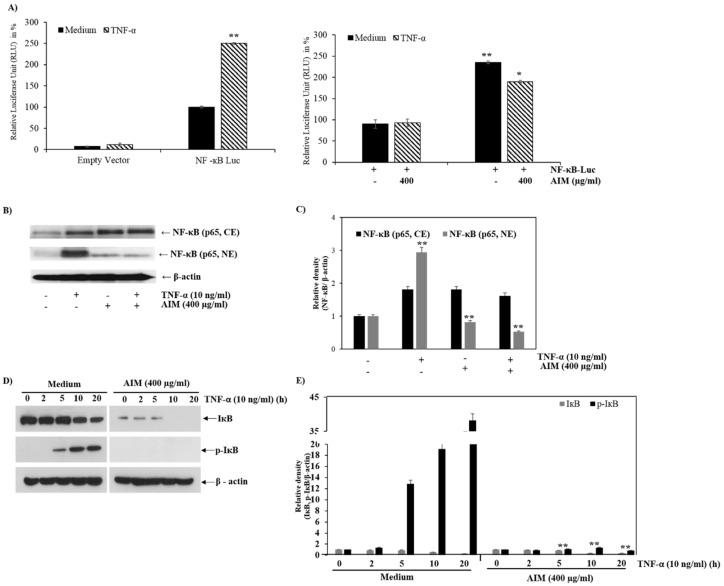
Effects of AIM on NF-κB and the IκBα phosphorylation. (**A**, Left) 1 μg of the NF-κB-luciferase empty vector were transfected into MCF-7 cells; (right) 24 h after transfection, the cells were treated with 10 ng/mL of TNF-α, with or without 1 h pre-treatment of AIM (400 μg/mL). The luciferase activity was represented in fold change and compared against the untreated group. (**B**) Repressive effects of AIM on TNF-α-induced NF-κB translocation from the cytoplasm to nucleus. The cells treated with 10 ng/mL of TNF-α pretreated with or without 400 μg/mL of AIM for 1 h were taken for western blot analysis. Nuclear (NE) and cytoplasmic extracts (CE) isolated from the total lysates were resolved in SDS-polyacrylamide gels. (**C**) Densitometry analysis of the data in western blot analysis by ImageJ software. The values were normalized against β-actin. (**D**) Inhibitory effects of AIM on TNF-α induced IκBα phosphorylation. A total of 30 μg of protein lysates from the indicated treatment cells were used for western blot analysis. (**E**) Densitometry analysis of the data in western blot analysis by ImageJ software. The values were normalized against β-actin. ** *p* < 0.01 vs. the control group.

**Figure 6 molecules-25-02396-f006:**
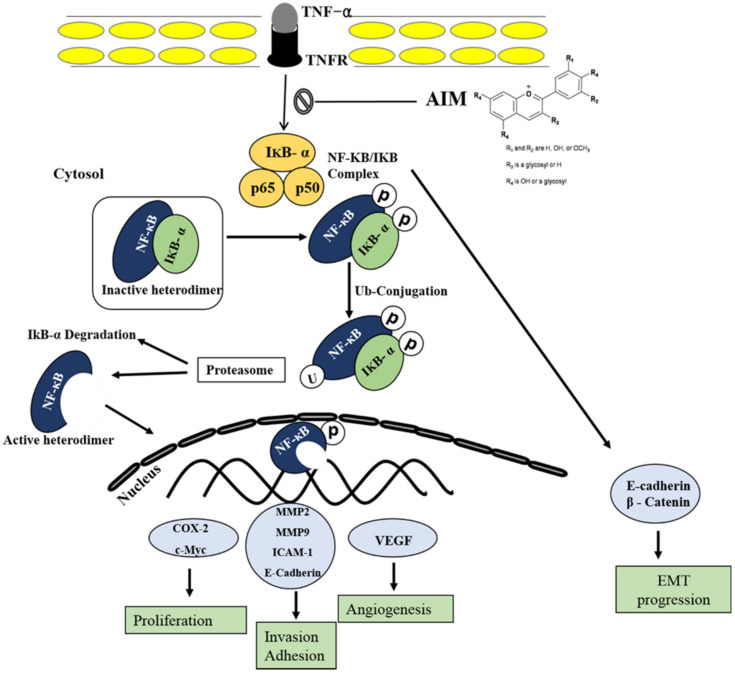
Schematic representation of AIM on TNF-α induced effect on MCF-7 breast cancer cells. Following activation of the NF-κB pathways, AIM inhibited TNF-α-induced NF-κB activities and NF-κB-regulated proteins involved in cancer cell proliferation, invasion, and angiogenesis. Taken together, these data suggest that AIM may act as a TNF-α inhibitor by suppressing NF-κB pathways on human breast cancer.

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
