# Peer review of "Pretreatment of Anthocyanin from the Fruit of Vitis coignetiae Pulliat Acts as a Potent Inhibitor of TNF-α Effect by Inhibiting NF-κB-Regulated Genes in Human Breast Cancer Cells"

_molecules, 2020, doi:10.3390/molecules25102396_

Round 1
Reviewer 1 Report
The manuscript by Paramanantham et al describes the anti-metastatic properties of an anthocyanin-rich extract from Vitis coignetiae Pulliat (AIM) in MCF7 cancer cells. Data indicate that the anthocyanin-rich exerts its anti-metastatic activity by inhibiting the TNFa-induced NF-kB activation of target genes/proteins involved in invasion, adhesion, and angiogenesis. However, the effect of AIM in suppressing the phosphorylation of IkBa in TNFa-induced MCF-7 cancer cells (Fig. 6a) appears not be evident, since a complete suppression of total IkBa protein is observed. Therefore, the authors should change their conclusions. Other specific comments are detailed below.
Line 35. … and involve anti-metastatic effects (this part should be rephrased).
Line 91. Define ECs
Figures Panels should be all indicated with letters and not subheadings (i, ii etc)
Figure 3A. Low quality panel. Please, use high quality images
Line 137 suppression (speling error)
Line 141 COX-2 and C-myc ARE overexpressed in different types of cancers and MEDIATE cancer cell proliferation
Line 144 Define EMT
Line 149. In many cases (e.g. beta-catenin, ICAM-1, surbivin, VEGF), TNFa does not increase protein expression over CNT. This sentence should be modified accordingly and this point should be commented in the discussion. Statistical significance compared to control should be also indicated in Fig.4, panel B
Line 174. Actually, this conclusion is unjustified. Since AIM completely suppress IkBa total protein level, the lack of IkBa phosphorylation cannot be demonstrated.
Line 236 the last sentence is incomplete
Line 277 … WERE treated with the indicated concentration
Lines 280-281. The description of the invasion assay is unclear and should be better explained. Which parameter has been evaluated?
Author Response
Response to Reviewer 1 Comments
Point 1: The manuscript by Paramanantham et al describes the anti-metastatic properties of an anthocyanin-rich extract from Vitis coignetiae Pulliat (AIM) in MCF7 cancer cells. Data indicate that the anthocyanin-rich exerts its anti-metastatic activity by inhibiting the TNFa-induced NF-kB activation of target genes/proteins involved in invasion, adhesion, and angiogenesis. However, the effect of AIM in suppressing the phosphorylation of IkBa in TNFa-induced MCF-7 cancer cells (Fig. 6a) appears not be evident, since a complete suppression of total IkBa protein is observed. Therefore, the authors should change their conclusions. Other specific comments are detailed below
Response 1:Thank you for your comments. We really agree with your opinion. It looks like that they are mismatched findings. We do not know why AIM far much increased the cytoplasmic NF-kB and total NF-kB. But we focus on the TNF-α-treated condition. In TNF-α-treated condition, AIMs significantly inhibited translocation of NF-kB from cytoplasm to nucleus compared to TNF-α alone.
From the above findings and interpretation, we conclude, that the anthocyanins isolated from fruits of Vitis coignetiae Pulliat acts as an inhibitor of TNF-α induced NF-κB activation.
If you have a better suggestion, please give us a more specific suggestion.
Point 2: Line 35. … and involve anti-metastatic effects (this part should be rephrased).
Response 2:Thank you for your comments. According to your suggestion, we have corrected it.
In the revised manuscript, we corrected it as follows: thus have anti-metastatic activities in MCF-7 breast cancer cells.
Point 3: Line 91. Define Ecs
Response 3:Thank you for your comments. According to your suggestion, we have defined it. Human umbilical vein endothelial cells (ECs).
Point 4: Figures Panels should be all indicated with letters and not subheadings (i, ii etc)
Response 4: Thank you for your comments. According to your suggestion, we have corrected it. In the revised manuscript, instead of subheading, we used upper, and lower in figure legends.
Point 5: Figure 3A. Low quality panel. Please, use high quality images.
Response 5:Thank you for your comments. According to your suggestion, we have corrected it.
Point 6: Line 137 suppression (speling error)
Response 6:Thank you for your comments. According to your suggestion, we have corrected it.
Point 7: Line 141 COX-2 and C-myc ARE overexpressed in different types of cancers and MEDIATE cancer cell proliferation
Response 7:Thank you for your comments. According to your suggestion, we have corrected it.
Point 8: Line 144 Define EMT
Response 8:Thank you for your comments. According to your suggestion, we have defined it. Epithelial-Mesenchymal Transition (EMT).
Point 9: Line 149. In many cases (e.g. beta-catenin, ICAM-1, surbivin, VEGF), TNFa does not increase protein expression over CNT. This sentence should be modified accordingly and this point should be commented in the discussion. Statistical significance compared to control should be also indicated in Fig.4, panel B
Response 9: Thank you for your comments. According to your suggestion, we have corrected it.
TNF-α increased the expression of beta-catenin, ICAM-1, survivin, VEGF in comparison with the AIM treated group. Statistical significance compared to control is also included in Fig.4, panel B
Point 10: Line 174. Actually, this conclusion is unjustified. Since AIM completely suppress IkBa total protein level, the lack of IkBa phosphorylation cannot be demonstrated.
Response 10: Thank you for your comments. We really agree your opinion.
With AIM’s effects on NF-kB in control cells, we do not know exactly how AIM reduced IkBa total protein level with the lack of IkBa phosphorylation.
So we added it as a limitation in the discussion part as follows: In this study, we did not answer why AIM far much increased the cytoplasmic NF-kB and total NF-kB, and how AIM reduced IkBa total protein level with the lack of IkBa phosphorylation. Regarding reduction of IkBa with the lack of IkBa phosphorylation, IκB Kinase-independent IκBα degradation was reported [1]. Regarding these question, further study is warranted.
Point 11: Line 236 the last sentence is incomplete
Response 11: Thank you for your comments. According to your suggestion, we have corrected it.
Point 14: Line 277 … WERE treated with the indicated concentration
Response 14:Thank you for your comments. According to your suggestion, we have corrected it.
Point 15: Lines 280-281. The description of the invasion assay is unclear and should be better explained. Which parameter has been evaluated?
Response 15: Thank you for your comments. According to your suggestion, we have corrected it. We evaluated the number of invasive cells after the treatment of AIM with or without TNF-α stimulation.
We modified the following sentences in the material and methods section
- Transwell assay was performed to analyze the effect of AIM and TNF-α on cell invasion. The upper chamber of the Boyden chamber was coated with Matrigel (0.5 mg/ml) (BD Biosciences, San Jose, CA) and incubated at 37 °C for 4 h.” To”Transwell assay was performed to analyze the effect of AIM on cell invasion with or without TNF-α treatment The upper chamber of the Boyden chamber was coated with Matrigel (0.5 mg/ml) (BD Biosciences, San Jose, CA) and incubated at 37 °C for 4 h.” Line no: 294-295
- From “MCF-7 cells were cultured overnight in a serum free media. 5 x 104 cells/well of MCF-7 cells were added to the upper chamber with the serum free media with and without AIM (400 μg/ml) treatment followed by the TNF-α” To “MCF-7 cells were cultured overnight in a serum free media. 5 x 104 cells/well of MCF-7 cells were added to the upper chamber after solidification of Matrigel, with the serum free media with and without AIM (400 μg/ml) treatment followed by the TNF-α (10 ng/ml) stimulation.” Line no :298-299
- From”After staining with DAPI, the cells were counted manually using Fluorescent microscope.”To ”After staining with DAPI, the invasive cells were counted manually using Fluorescent microscope.” Line no:303
Reviewer 2 Report
Paramanantham et al describe the effects of anthocyanins isolated from Meoru (AIM) on the growth and NF-kappa related proteins on MCF-7 cells. The paper is well written and the data is interesting. However, the growth inhibitory effect is found at high concentrations only. The authors describe some of the limitations of this study. It is not clear as to the effect of this agent on normal cells. In addition, the study is conducted on one breast cancer cell line only and hence the general applicability is questionable. The authors may conduct additional experiments or describe these limitations also.
Author Response
Response to Reviewer 2 Comments
Point 1: Paramanantham et al describe the effects of anthocyanins isolated from Meoru (AIM) on the growth and NF-kappa related proteins on MCF-7 cells. The paper is well written and the data is interesting. However, the growth inhibitory effect is found at high concentrations only. The authors describe some of the limitations of this study. It is not clear as to the effect of this agent on normal cells. In addition, the study is conducted on one breast cancer cell line only and hence the general applicability is questionable. The authors may conduct additional experiments or describe these limitations also.
Response 1: Thank you for your comments. According to your suggestion, we have corrected it.
We added it as a limitation in the discussions part as follows:
Regarding the concentration of the AIM used in this study, it might seem high for in-vivo studies and in normal cells, but in the adhesion study, no toxicity was observed to ECs and we have previously demonstrated its effects at several concentrations of AIM in-vivo studies [2].
With reference.
- Tergaonkar V, Bottero V, Ikawa M, Li Q, Verma IM: IkappaB kinase-independent IkappaBalpha degradation pathway: functional NF-kappaB activity and implications for cancer therapy. Mol Cell Biol 2003, 23(22):8070-8083.
- Lee YK, Lee WS, Kim GS, Park OJ: Anthocyanins are novel AMPKalpha1 stimulators that suppress tumor growth by inhibiting mTOR phosphorylation. Oncology reports 2010, 24(6):1471-1477.